# Cognitive Safety and Outcomes of Pharmacological Management in Heart Failure: A Systematic Review

**DOI:** 10.3390/ph18111671

**Published:** 2025-11-05

**Authors:** Wiktoria Balcerzak, Gabriela Poczatek, Agnieszka Gorzkowska, Anna Blach, Michal Jurkiewicz, Anetta Lasek-Bal

**Affiliations:** 1Department of Neurology, Upper-Silesian Medical Center of the Medical University of Silesia, 40-055 Katowice, Poland; agorzkowska@sum.edu.pl (A.G.); abal@sum.edu.pl (A.L.-B.); 2Doctoral School of Medical and Health Sciences, Medical University of Silesia, 40-055 Katowice, Poland; gabrielaanielapoczatek@gmail.com; 3Department of Neurology, School of Health Sciences, Medical University of Silesia, 40-055 Katowice, Poland; 4Department of Cardiology and Structural Heart Diseases, Medical University of Silesia, 40-055 Katowice, Poland; ablach@sum.edu.pl; 5Nuclear Medicine Department, Voxel Diagnostic Center, 40-514 Katowice, Poland; 6Students’ Scientific Society, Department of Neurology, School of Health Sciences, Medical University of Silesia, 40-055 Katowice, Poland; s81184@365.sum.edu.pl

**Keywords:** heart failure, cognitive impairment, pharmacological treatment

## Abstract

**Background/Objectives**: Cognitive impairment (CI) is a common complication of heart failure (HF) that undermines self-care, adherence, and outcomes. This systematic review assessed the cognitive effects of pharmacological HF management and related cardiovascular therapies, identifying potential risks and benefits. **Methods**: Following PRISMA 2020, we searched PubMed, EMBASE, and Scopus for articles published in English between 1 January 2010 and 31 January 2025 (last search 31 January 2025). We included RCTs and cohort studies in adults with HF or high cardiovascular risk that reported cognitive outcomes. Non-pharmacological interventions, studies without relevant cognitive endpoints, and non-original research were excluded. Risk of bias was assessed using RoB 2 for RCTs and the Newcastle–Ottawa Scale for observational studies. Due to heterogeneity in study designs and cognitive measures, results were synthesized qualitatively without meta-analysis. **Results**: Of 530 records screened, 11 studies encompassing 58,190 participants met the inclusion criteria. Intensive blood pressure (BP) lowering was consistently associated with reduced risk of mild cognitive impairment or dementia compared with standard BP control. In HF populations, sacubitril/valsartan showed no adverse cognitive effects versus other RAAS inhibitors. Across RCTs and observational studies, β-blockers, ACE inhibitors, ARBs, diuretics, and statins showed no evidence of significant cognitive impairment. Comparisons of anticoagulants (dabigatran vs. warfarin; warfarin vs. aspirin) revealed no differences in cognitive trajectories, while optimized medical therapy was associated with parallel improvements in cognitive scores, left ventricular function, and renal parameters. **Limitations**: Evidence remains constrained by heterogeneity in study design and cognitive assessment tools (often brief screening instruments), inconsistent reporting, and generally short follow-up durations, which may obscure subtle or long-term effects. **Conclusions**: Contemporary pharmacological therapies for HF appear cognitively safe. Intensive BP control may confer cognitive benefit in high-risk populations, while guideline-directed treatments show no consistent evidence of cognitive harm. Optimized medical therapy may even support cognitive improvement alongside cardiac and renal recovery. Routine cognitive assessment should be integrated into HF care to support individualized management.

## 1. Introduction

HEART failure (HF) affects an estimated 56 million individuals globally, representing a significant public health issue [1,2]. HF is a heterogeneous clinical syndrome with both cardiac and non-cardiac causes, among which coronary heart disease (CHD), arrhythmias, valvular heart disease, hypertension, diabetes, and obesity are the most common underlying conditions. Its increasing prevalence is largely due to an ageing population, improved causal treatments, and better survival rates.

According to the European Society of Cardiology (ESC) guidelines, chronic HF is classified by left ventricular ejection fraction (LVEF) into three subgroups: HF with preserved ejection fraction (HFpEF, LVEF ≥ 50%), HF with mildly reduced ejection fraction (HFmrEF, LVEF 41–49%), and HF with reduced ejection fraction (HFrEF, LVEF ≤ 40%) [3,4] (Table 1).

In addition to pharmacological treatments, heart failure management includes palliative care, metabolic interventions, and the use of implantable devices, all of which may also impact cognitive outcomes. Although these approaches are not standard chronic HF therapies, they offer valuable insights into the comprehensive care of heart failure populations, particularly in advanced stages of the disease [3,5,6].

Due to its high metabolic demands, the brain is critically dependent on adequate blood flow. Any reduction in cardiac output can lead to decreased cerebral perfusion, resulting in hypoxia and inadequate delivery of nutrients and oxygen, as well as neurovascular dysfunction. Structural brain abnormalities associated with HF—including brain atrophy, white matter lesions, and grey matter loss in regions critical for cognition, such as the frontal cortex and hippocampus—have been linked to cognitive impairment (CI), memory loss, and an increased risk of dementia [5].

Moreover, the relationship between heart and brain is bidirectional: the brain regulates cardiac function via the autonomic nervous system, and disruptions in this regulatory loop can adversely affect both organs, which also share common risk factors [6,7]. CI has been identified as a significant barrier to effective disease management, reducing patients’ ability to self-care, adhere to treatment regimens, and maintain autonomy. This, in turn, exacerbates the challenges of managing chronic disease and increases healthcare burdens.

A robust association has been demonstrated between HF and cognitive dysfunction. Patients with HF show reduced cognitive abilities and are at a higher risk of developing dementia compared to the general population [8,9]. Cognitive impairment is common in HF, with prevalence estimates ranging from 20% to 80% [8]. Its etiology is multifactorial, reflecting a complex interplay of risk factors and mechanisms that include primary hemodynamic abnormalities, structural neuronal changes, as well as neurohormonal and inflammatory processes associated with HF [10].

A variety of neuropsychological tests are available to clinicians and other healthcare professionals for assessing cognitive function in HF patients. These tests are very diverse and allow the tool to be selected for the patient’s specific needs. However, it is important to note that most of these tests require appropriately trained staff and are time-consuming. Currently, there is no consensus on the optimal screening tool for cognitive impairment in HF, and most studies rely on brief screening measures. The Mini-Mental State Examination (MMSE) and the Montreal Cognitive Assessment (MoCA) are the most widely used instruments for this purpose [11].

Tests generally show that patients with HF have reduced overall cognitive performance compared to healthy individuals, with deficits described in several domains, including executive function, psychomotor speed, and memory [12]. As one of the most important cognitive functions, episodic memory for specific personal events and experiences has been shown to decline gradually in patients with HF [13]. Additionally, impairments in initial learning and delayed recall of information have been described in the literature [14]. Executive function difficulties (attention, planning, problem solving, complex decision–making) have a major impact on patients’ daily functioning. These difficulties are prevalent in frontotemporal and vascular dementia and are also found in patients with HF [15].

Monitoring and improving cognitive function in HF is crucial for effective disease management [16]. Preserved memory and executive functioning are especially essential for recognizing worsening symptoms, adhering to prescribed treatments, making lifestyle modifications, and attending regular follow-up visits. Even mild deficits in these areas can significantly reduce adherence to self-care practices, negatively affecting treatment outcomes and quality of life [17]. Given this impact, cognitive impairment should be carefully considered in clinical decision-making [18].

Beyond these hemodynamic and structural contributors, pharmacological treatment itself may influence brain function through direct or indirect neurobiological pathways. The pharmacological management of HF may interact with these brain–heart mechanisms through several routes. β-blockers, depending on their lipophilicity and β_1_-selectivity, can cross the blood–brain barrier to varying degrees and modulate central noradrenergic signalling and cerebral perfusion. Evidence indicates that β-blockers differ in their cognitive profiles depending on lipophilicity and receptor selectivity: non-selective lipophilic agents (e.g., propranolol) may transiently affect memory processes, while selective β_1_-blockers tend to preserve or even support cognitive function in the setting of chronic sympathetic activation [19]. Likewise, modulators of the renin–angiotensin–aldosterone system may exert neuroprotective effects by reducing neuroinflammation, oxidative stress, and endothelial dysfunction, as well as by improving cerebral blood flow and promoting neurogenesis [20]. Diuretics indirectly affect cognition through volume status and electrolyte balance. Hypovolemia or hyponatremia can impair cognitive function; thus, careful titration is crucial in elderly HF patients [21]. Understanding these neurobiological links is essential for interpreting how pharmacological therapy may influence cognitive trajectories in HF.

Given the significant burden of cognitive impairment in HF patients, it is crucial to evaluate the influence of contemporary pharmacological therapies on their cognitive trajectories. This systematic review aimed to synthesize current evidence on the cognitive safety and potential neurocognitive effects of pharmacological treatments used in heart failure.

## 2. Materials and Methods

### 2.1. Literature Review

A systematic literature review was conducted in accordance with the PRISMA 2020 statement (Preferred Reporting Items for Systematic Reviews and Meta-Analyses) [22] between 1 December 2024 and 31 January 2025. A comprehensive search of PubMed, EMBASE, and Scopus was performed for articles published in English between 1 January 2010 and 31 January 2025.

Two complementary search strategies were applied: a primary strategy (heart failure AND cognition AND drug therapy), which identified 374 records, and a supplementary strategy (heart failure AND cognition AND drug therapy AND cognitive test), designed to capture studies in which cognitive terminology was limited to the descriptor “cognitive test”. The supplementary search yielded 156 additional records. After deduplication and screening according to predefined inclusion criteria, eight studies were included from the primary search and three from the supplementary search, resulting in a total of 11 studies included in the review.

Reference lists of included articles were manually screened to identify additional eligible studies. Grey literature and non-peer-reviewed sources were excluded.

The review protocol was registered in PROSPERO (CRD420251156308).

### 2.2. Eligibility Criteria

Eligibility was determined using a PICOS framework adapted to the scope of this review:Population: Adult patients (≥18 years) with diagnosed heart failure or at high cardiovascular risk.Intervention: Pharmacological treatment for heart failure or cardiovascular conditions (e.g., antihypertensives, RAAS inhibitors, beta-blockers, sacubitril/valsartan, statins, or other agents).Comparison: Usual care, placebo, or alternative pharmacological regimens.Outcomes: Cognitive function measured using standardized tools (e.g., MMSE, MoCA) or clinical cognitive endpoints (e.g., diagnosis of dementia, cognitive decline).Study design: Randomized controlled trials (RCTs), observational cohort studies.

Studies were excluded if they:Did not report cognitive outcomes;Focused exclusively on non-pharmacological interventions;Included populations not relevant to heart failure or cardiovascular risk;Lacked sufficient detail on patient characteristics (e.g., no clear information on post-myocardial infarction status or cardiovascular comorbidities);Publication types other than original research (e.g., reviews, meta-analyses, editorials, commentaries, case reports);Studies published before 2010.

### 2.3. Data Extraction

Three reviewers screened titles and abstracts independently, with disagreements resolved through discussion and consensus. Extracted variables included study design, sample size, population characteristics, pharmacological interventions, cognitive assessment methods, and main findings related to cognitive outcomes.

### 2.4. Synthesis, Risk of Bias, and Certainty of Evidence

Given the heterogeneity of study designs and outcome measures, no formal meta-analysis was performed. Results were synthesized qualitatively and presented in tabular form to highlight study characteristics, interventions, cognitive assessment tools, and key findings. No quantitative pooling was performed due to heterogeneity in cognitive measures.

Risk of bias was assessed independently by three reviewers. Randomized controlled trials were evaluated using the Cochrane Risk of Bias 2 (RoB 2) tool, and observational cohort studies were assessed using the Newcastle–Ottawa Scale (NOS). Discrepancies were resolved by consensus.

The certainty of evidence for each outcome was assessed using the GRADE framework, considering risk of bias, inconsistency, indirectness, imprecision, and reporting bias. Certainty was classified as high, moderate, low, or very low.

## 3. Results

### 3.1. Study Selection

A total of 544 records were identified through database searches. After removing 96 duplicates, 448 records remained for title and abstract screening. Of these, 223 were excluded based on titles and abstracts. A total of 139 full-text reports were sought for retrieval, with 2 not successfully retrieved. Ultimately, 137 full-text articles were assessed for eligibility. After applying inclusion and exclusion criteria, 11 studies were included in the qualitative synthesis. The study selection process is illustrated in the PRISMA 2020 flow diagram (Figure 1). Further details are provided in the Appendix A.

### 3.2. Study Characteristics

The 11 included studies comprised both randomized controlled trials and observational cohort studies examining the relationship between pharmacological treatment for HF (or cardiovascular conditions) and cognitive outcomes.

Sample sizes varied considerably, ranging from small single-centre studies with fewer than 50 participants to large multicentre randomized trials enrolling several thousand patients. Populations studied included individuals with HFrEF, HFpEF, hypertension with cardiovascular risk factors, atrial fibrillation, and mixed cardiovascular cohorts.

Pharmacological interventions evaluated encompassed a broad range of agents, including RAAS inhibitors (ACEIs, ARBs, and sacubitril/valsartan), beta-blockers, diuretics, statins, and anticoagulants, as well as antihypertensive and adjunctive therapies such as intensive blood pressure–lowering regimens, pioglitazone, and morphine.

Cognitive outcomes were assessed primarily using brief screening tools such as the MMSE and MoCA, although some studies used broader neuropsychological batteries or clinical dementia diagnoses.

The key characteristics and findings of the included studies are summarized in Table 2.

### 3.3. Synthesis of Findings

Across the included studies, intensive blood pressure lowering was associated with a reduced risk of mild cognitive impairment or dementia in high-risk populations. Randomized evidence (e.g., SPRINT analyses) demonstrated a consistent cognitive benefit for more intensive antihypertensive regimens compared with standard care, particularly among patients with lower cardiac biomarker levels. Regarding heart failure–specific pharmacotherapies, in HFpEF, sacubitril/valsartan showed no adverse cognitive effects versus comparator RAAS inhibition (PARAGON-HF), and observational data further confirmed its cognitive safety profile. Studies evaluating other cardiovascular medications—β-blockers, ACE inhibitors, ARBs, diuretics, and statins—generally did not demonstrate significant detrimental effects on cognitive function. Additionally, long-term optimized medical therapy (OMT) was associated with parallel improvements in cognitive scores, left ventricular function, and renal parameters, suggesting potential global benefits of comprehensive pharmacological optimization in elderly heart failure patients. However, most studies relied on brief screening tools, which may not capture subtle or domain-specific cognitive changes.

### 3.4. Risk of Bias and Certainty of Evidence

The overall methodological quality of the included studies was moderate to low. Most RCTs were not specifically designed to assess cognitive outcomes, and cognitive assessments were often limited to brief screening tools such as the MMSE or MoCA. Follow-up durations were frequently short, and statistical reporting was heterogeneous.

Formal assessment confirmed these limitations. According to the Cochrane RoB 2 tool, RCTs were judged to be at low to moderate risk of bias, with common concerns regarding blinding of outcome assessors and selective reporting of secondary cognitive endpoints. Observational studies, evaluated using the Newcastle–Ottawa Scale, generally scored 6–7 stars, reflecting moderate quality with risk of confounding and incomplete adjustment for comorbidities (Table 3).

The certainty of evidence, assessed using the GRADE framework, varied across interventions. Evidence for intensive blood pressure lowering was rated as moderate certainty, whereas sacubitril/valsartan and other guideline-directed HF therapies were graded as low certainty. Evidence regarding anticoagulants and statins was judged very low. No consistent signal of harm was identified across other pharmacological classes (Table 4).

Formal assessment of reporting bias (e.g., funnel plots, Egger’s test) was not feasible due to the small number of studies per outcome; potential for selective reporting is acknowledged.

## 4. Discussion

This systematic review indicates that contemporary pharmacological therapies for HF are generally cognitively safe, with no consistent evidence of harm and, in some cases, potential to improve outcomes. Intensive BP-lowering strategies were consistently associated with reduced risk of cognitive decline. Large randomized trials, such as those based on the SPRINT analysis, showed that targeting lower systolic BP (<120 mmHg versus <140 mmHg) reduced the risk of MCI or dementia diagnoses, including in patients with elevated cardiac biomarkers (High-Sensitivity Troponin T, NT-proBNP) [23]. These findings support the hypothesis that optimizing systemic hemodynamics can mitigate cerebral hypoperfusion, a key mechanism underlying HF-related cognitive decline.

Our results are consistent with recent systematic reviews examining the cognitive safety of contemporary heart failure therapies. Although both works address similar research questions, they differ in scope and analytical focus. Jahangiri et al. [34] systematically evaluated the cognitive effects of guideline-directed heart failure therapies, emphasizing sacubitril/valsartan and emerging drug classes such as SGLT2 inhibitors. In contrast, our review takes a broader clinical perspective—covering not only GDMT but also other cardiovascular and supportive pharmacotherapies relevant to heart failure populations (e.g., anticoagulants, statins, pioglitazone, morphine). Furthermore, our analysis integrates mechanistic insights into the neurobiological pathways by which β-blockers and RAAS modulators may influence cognition, applies GRADE methodology for evidence certainty, and provides practical recommendations on cognitive monitoring in HF care. These complementary approaches collectively enhance understanding of cognitive safety and translational relevance of pharmacological management in heart failure.

Recent reports on SGLT2 inhibitors further contribute to this evolving discussion on the cognitive effects of modern HF therapies. Although no SGLT2 inhibitor trials met our inclusion criteria, emerging evidence outside HF populations provides preliminary insight into potential cognitive effects. In a prospective study of frail older adults with HFpEF and type 2 diabetes, empagliflozin therapy was associated with a significant short-term improvement in MoCA scores compared with baseline, suggesting a possible cognitive benefit through hemodynamic or metabolic mechanisms [35]. Conversely, a recent randomized controlled trial in patients with early Alzheimer’s disease found that dapagliflozin did not alter the primary neurochemical marker of neuronal integrity (N-acetylaspartate), although modest improvements were observed in metabolic parameters and Stroop test performance [36]. While these studies were not eligible for inclusion in our systematic review, they provide hypothesis-generating evidence supporting further investigation of SGLT2 inhibitors in HF populations using standardized, multidomain cognitive assessments and longer follow-up.

Evidence regarding digoxin and cognitive function in heart failure remains scarce and largely indirect. No randomized or prospective studies have specifically evaluated cognitive outcomes in patients treated with digoxin. Available data are limited to case reports and observational findings suggesting that cognitive disturbances—when present—are usually related to digoxin toxicity, renal dysfunction, or drug–drug interactions rather than therapeutic use. Thus, the cognitive safety of digoxin at standard dosages in heart failure remains uncertain and warrants systematic evaluation in future studies [37].

Together with these emerging data on SGLT2 inhibitors, our findings suggest that contemporary HF pharmacotherapy as a whole appears cognitively neutral or potentially beneficial. Among established GDMT agents, sacubitril/valsartan, β-blockers, and RAAS inhibitors have shown no consistent evidence of cognitive harm in either randomized or observational settings. Trials such as PARAGON-HF found no differences in cognitive outcomes, including MMSE scores and dementia events, between sacubitril/valsartan and valsartan. Despite theoretical concerns about neprilysin inhibition and amyloid-β accumulation, no indication of cognitive impairment emerged [24].

Other standard cardiovascular therapies also showed no evidence of harming cognition. Palau et al. investigated the effects of discontinuing versus continuing beta-blockers in patients with HFpEF and chronotropic incompetence, reporting no differences in MMSE or MoCA scores [25]. Observational data from Bratzke et al. also found no association between guideline-directed HF medications (β-blockers, ACE inhibitors, ARBs, diuretics, MRAs) and cognitive impairment in a cohort of recently hospitalized HF patients [27]. These findings suggest that concerns about drug-induced cognitive impairment (DICI) should not discourage the use of evidence-based HF therapies.

Recent studies further support these observations. The GIRAF trial compared dabigatran with warfarin over 24 months in older adults with atrial fibrillation and found no significant differences in MMSE or MoCA performance after adjustment for multiple comparisons, suggesting that direct oral anticoagulants do not confer measurable cognitive advantage beyond stroke prevention [31]. Similarly, De Vecchis et al. (2019) reported comparable cognitive performance among HF patients treated with sacubitril/valsartan versus standard therapy, reinforcing the cognitive safety profile of ARNIs [33]. In a small prospective study, Osipova et al. (2023) [32] observed that 12-month optimized medical therapy (OMT) improved both MoCA and MMSE scores in elderly HF patients, in parallel with better LV ejection fraction and renal function. Although these findings should be interpreted cautiously due to limited sample sizes and potential confounding, they suggest that comprehensive pharmacological optimization may positively influence cognitive trajectories in HF [32].

Our review also included studies on non-standard or adjunctive pharmacotherapies. Hildreth et al. assessed pioglitazone in older adults with central obesity and MCI, finding improved insulin resistance but no cognitive benefit on MMSE scores [28]. Johnson et al. evaluated morphine for breathlessness in advanced HF (NYHA III/IV), observing no overall impairment in MoCA scores compared to placebo, except in a single patient with renal dysfunction [29]. Although these agents are not standard chronic HF therapies, these studies provide valuable safety insights for palliative or metabolic management in HF populations.

Notably, statin therapy showed no consistent cognitive effects. Large trials in older adults yielded equivocal results, suggesting that any cognitive impact—positive or negative—is likely minimal and difficult to detect in the short timeframes [26]. Similarly, warfarin versus aspirin in atrial fibrillation showed no clear cognitive difference beyond stroke prevention [30]. Overall, these findings suggest that standard cardiovascular therapies do not appear to worsen cognitive outcomes in HF patients, which is reassuring given the importance of these therapies for survival and symptom control.

Beyond evaluating the cognitive safety of pharmacological therapies, it is also essential to consider the prognostic role of cognition in HF. Cognitive impairment is increasingly recognized as an independent predictor of adverse outcomes, including rehospitalization, mortality, and poorer quality of life. Findings from our review suggest that optimal blood pressure control and adherence to guideline-directed medical therapy are compatible with preserved cognitive function and remain crucial for improving both cardiovascular and cognitive outcomes [23,27]. These observations highlight that cognitive status is closely tied to prognosis in HF and should be routinely assessed in clinical practice to support risk stratification, optimize adherence, and facilitate more individualized management strategies.

Given the high prevalence of cognitive impairment among HF patients—estimated at 20–80%—and its impact on self-care, adherence, and outcomes, these findings have important clinical implications. Current evidence suggests clinicians should not avoid guideline-directed HF therapies due to concerns about cognitive harm. Instead, optimizing HF management—including intensive BP control when appropriate—may help preserve both cardiovascular and cognitive health.

Cognitive impairment is increasingly recognized as an integral component of heart failure syndrome and should be routinely assessed throughout the course of treatment. Simple screening tools such as MoCA and MMSE remain practical for longitudinal use in clinical settings, despite their limited sensitivity. However, multidomain instruments that evaluate executive, memory, and attention domains (e.g., Trail Making Test, Digit Symbol Coding, verbal fluency) provide more comprehensive information and should be implemented in specialized or research contexts. Regular cognitive monitoring—particularly after therapy initiation or medication adjustment—facilitates early detection of decline, supports shared decision-making, and improves treatment adherence. Integration of cognitive assessment into multidisciplinary HF care pathways is therefore essential to ensure both cardiovascular and neurocognitive stability.

Although no pharmacological agents are currently approved specifically for cognitive impairment in HF, several therapeutic strategies may indirectly mitigate cognitive decline by optimizing cerebral perfusion and reducing neuroinflammation. Evidence suggests that strict blood pressure control, management of arrhythmias, and correction of anemia or hypoxia can prevent secondary cognitive deterioration [6,16,18]. Neuroprotective effects have also been observed with agents targeting the renin–angiotensin–aldosterone system, which may improve endothelial function, cerebral autoregulation, and oxidative balance [19,20]. Similarly, sodium–glucose cotransporter-2 inhibitors appear to exert vascular and metabolic benefits that may support cognitive resilience [35,36]. Beyond these mechanisms, adjunctive neuroprotective or nootropic therapies—such as citicoline, ginkgo biloba extract, or pramiracetam—have shown modest cognitive benefits in vascular and mixed-etiology cognitive impairment in smaller clinical trials [38,39,40]. Although their evidence base in HF-specific populations remains limited, their use may be considered on an individualized basis following neurologist consultation, particularly in patients with persistent or progressive cognitive symptoms despite optimized HF management. Non-pharmacological measures, including regular physical activity, cognitive training, and control of vascular risk factors (e.g., diabetes, dyslipidemia), remain important adjuncts to preserve cognitive function and overall clinical stability in this population [16,17].

Future research should prioritize the development and adoption of standardized, multidomain neuropsychological batteries to comprehensively assess memory, executive function, visuospatial skills, language, and verbal fluency. The comprehensive cognitive battery applied in the SPRINT trial provides a valuable model for capturing subtle changes in cognition among HF populations [23]. Such approaches will enable more precise evaluation of treatment effects and more individualized management strategies.

Overall, contemporary pharmacological therapies for heart failure appear cognitively safe, with no consistent evidence of adverse effects on cognitive function. Intensive blood pressure control may confer cognitive benefit in high-risk populations, while optimized medical therapy could support parallel improvements in cardiac, renal, and cognitive parameters. Certain drug classes—such as ARNIs and SGLT2 inhibitors—may offer potential cognitive benefits, though supporting data remain limited. Routine cognitive assessment should be integrated into heart failure management to facilitate early detection of decline, optimize adherence, and support personalized care planning.

### Limitations

Several limitations of the evidence base deserve attention. The included studies varied in design, populations, treatment regimens, and follow-up duration. Importantly, the variability in cognitive assessment methods across studies limits the interpretability and comparability of results. Most of the included trials relied on brief screening tools such as the MMSE or MoCA [24,25,27,28,29,30], which, while practical, have limited sensitivity for detecting subtle or domain-specific deficits. Some studies even used less rigorous methods, such as telephone questionnaires [26], further highlighting inconsistency in measurement. These limitations constrain our ability to draw firm conclusions about small or long-term cognitive effects of specific drug classes or combinations.

In addition, our search was restricted to English-language publications, which may have introduced language bias and resulted in the omission of relevant non-English studies.

Despite these limitations, this review provides an important synthesis of the current evidence and highlights key areas for future investigation.

## 5. Conclusions

This systematic review indicates that contemporary pharmacological therapies for heart failure are generally cognitively safe. Intensive blood pressure lowering was consistently associated with a reduced risk of cognitive decline, supporting its role in mitigating neurovascular injury in high-risk patients. Sacubitril/valsartan demonstrated cognitive safety comparable to that of standard RAAS inhibitors in HFpEF populations, and other guideline-directed agents—including β-blockers, diuretics, and statins—showed no evidence of clinically meaningful cognitive harm.

Given the high prevalence and prognostic importance of cognitive impairment in heart failure, clinicians should routinely assess cognitive function and incorporate it into treatment planning. Certain drug classes, such as ARNIs and SGLT2 inhibitors, may confer cognitive benefits; however, the supporting evidence remains limited and requires confirmation in future large-scale studies employing standardized, multidomain cognitive assessments and extended follow-up.

Routine cognitive evaluation should be integrated into heart failure management to facilitate early detection of decline, improve treatment adherence, and support individualized care strategies.

## Figures and Tables

**Figure 1 pharmaceuticals-18-01671-f001:**
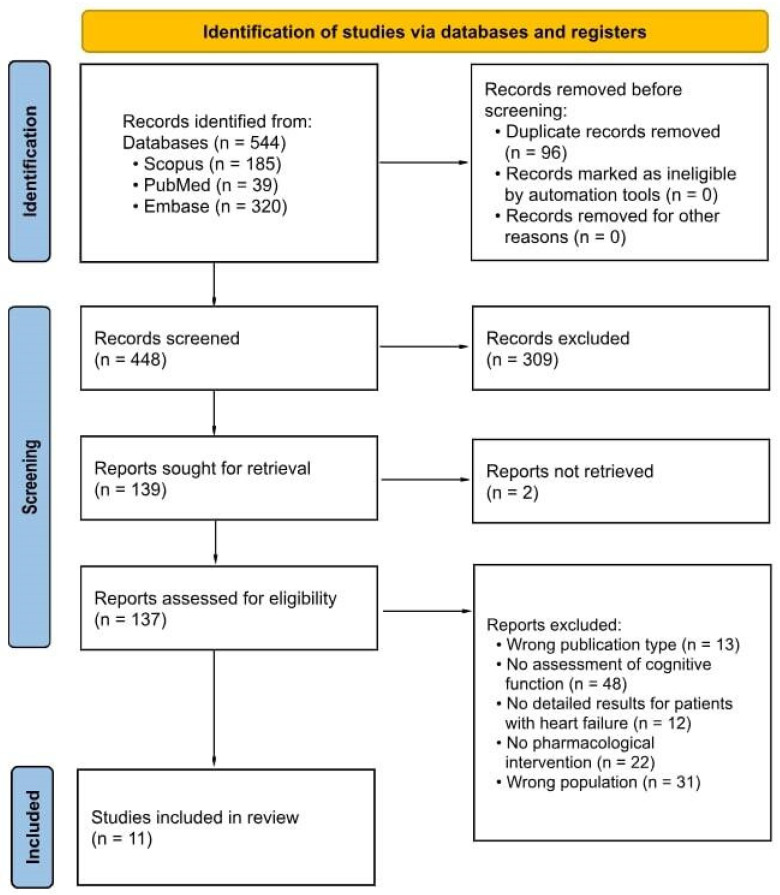
PRISMA 2020 flow diagram for study selection.

**Table 1 pharmaceuticals-18-01671-t001:** ESC classification of chronic heart failure by LVEF.

HF Type	HFpEF	HFmrEF	HFrEF
LVEF	≥50%	41–49%	≤40%

Abbreviations: HF—heart failure; HFpEF—heart failure with preserved ejection fraction; HFmrEF—heart failure with mildly reduced ejection fraction; HFrEF—heart failure with reduced ejection fraction; LVEF—left ventricular ejection fraction.

**Table 2 pharmaceuticals-18-01671-t002:** Summary of studies on cardiovascular drug treatment and cognitive outcomes.

Study (Year, Trial Acronym)	Participants (n)	Diagnosis/Population	Intervention	Cognitive Tools	Main Results (Effect Size, *p*-Value)	Key Findings
Haney et al. (2024, SPRINT) [23]	8111	Hypertension with cardiac biomarkers (Hs-cTnT, NT-proBNP), categorized into three groups based on biomarker levels (low, intermediate, high)	Intensive vs. Standard BP Control	MoCA, DSC	In the low biomarker group, intensive BP control significantly reduced the risk of MCI/dementia (HR 0.64; 95% CI 0.50–0.81; *p* < 0.05). Risk reduction was similar across all groups, but the greatest impact on cognitive decline prevention was seen in the low biomarker group (Interaction *p* = 0.02).	Intensive BP control reduces cognitive decline risk among those with lower cardiac biomarker levels (no overall harm).
Dewan et al. (2024, PARAGON-HF) [24]	2895	HFpEF	Sacubitril/Valsartan vs. Valsartan	MMSE	Change in MMSE at 96 weeks: sac/val −0.05 ± 0.07, valsartan −0.04 ± 0.07; between-group difference −0.01 (95% CI −0.20 to 0.19), *p* = 0.95.	No significant difference in cognitive change (MMSE) between sacubitril/valsartan and valsartan in HFpEF.
Palau et al. (2021) [25]	52	HFpEF + chronotropic incompetence	β-blocker Withdrawal vs. Continuation	MoCA, MMSE	Baseline MMSE ≈ 27.4 ± 1.9 vs. 27.5 ± 2.1 (*p* = 0.84); change +0.2 vs. +0.1 (*p* = 0.68); MoCA change +0.3 vs. +0.2 (*p* = 0.76).	In this small RCT, stopping β-blockers did not worsen (or improve) cognitive scores; cardiac functional capacity improved (reported elsewhere).
Offer et al. (2019) [26]	45,029	Vascular disease or diabetes (HPS/SEARCH/HPS2-THRIVE trials)	Statin therapy vs. Placebo/Usual care	TICS-m, CANTAB, MMSE, CDR	Expected cognitive ageing reduction ~0.15 years over 5 years with statins (95% CI 0.11–0.19); change not statistically detectable in trial (*p* > 0.05).	Statin therapy provided cardiovascular benefit, but any cognitive gains were very small (≈0.15-year delay) and not detectable in short-term trials.
Bratzke et al. (2016) [27]	612	Recently hospitalized HF patients	β-blockers, ACEIs/ARBs, diuretics, aldosterone antagonists (any)	Mini-Cog	Adjusted OR for cognitive impairment with GDMT: 0.84 (95% CI 0.51–1.38, *p* = 0.49).	Prescription of HF therapies did not correlate with worse cognition; no evidence that these medications cause cognitive decline.
Hildreth et al. (2015) [28]	78	Older adults with central obesity + MCI	Pioglitazone vs. Exercise vs. Control	MMSE	Baseline MMSE ~27.2 ± 2.1 (*p* = 0.69); change over 6 months: +0.2 vs. +0.2 vs. +0.1 (*p* = 0.98).	Pioglitazone improved insulin resistance (*p* = 0.002) but showed no cognitive benefit on MMSE.
Johnson et al. (2019) [29]	45	HF (NYHA III/IV) with chronic breathlessness	Morphine 20 mg vs. Placebo (for breathlessness)	MoCA	Baseline MoCA: ~24.2 ± 4.4 vs. 24.0 ± 5.0 (*p* = 0.89); change over 12 weeks: +0.2 vs. +0.4 (*p* = 0.85).	No significant cognitive difference with morphine use; single case of delirium with renal dysfunction.
Mavaddat et al. (2014) [30]	973	Atrial fibrillation, age ≥ 75	Warfarin (INR 2–3) vs. Aspirin (75 mg)	MMSE	Mean MMSE at 33 mo: warfarin higher by 0.49 points (95% CI −0.01 to 0.98) vs. aspirin (after imputation); difference not statistically significant.	No significant cognitive advantage of warfarin over aspirin beyond stroke prevention (Warfarin did not meaningfully slow cognitive decline in the first 33 months).
Caramelli et al. (2022, GIRAF) [31]	200	Older adults (≥70 years) with atrial fibrillation or flutter, CHA_2_DS_2_-VASc ≥ 2, no prior stroke or dementia at baseline	Dabigatran (110 mg or 150 mg twice daily) vs. Warfarin (dose adjusted to INR 2.0–3.0) for 24 months	MMSE, MoCA, NTB, CGNT	No significant differences between groups in most cognitive measures after 2 years. MMSE: Mean difference = −0.12 (95% CI −0.88 to 0.63; *p* = 0.75) MoCA: Mean difference = −0.96 (95% CI −1.80 to −0.13; *p* = 0.02; adjusted *p* = 0.08) NTB: Mean difference = 0.05 (95% CI −0.07 to 0.18; *p* = 0.40) CGNT: Mean difference = −0.15 (95% CI −0.30 to 0.01; *p* = 0.06)	Neither dabigatran nor warfarin showed superiority in cognitive outcomes after 2 years. While unadjusted MoCA results favoured warfarin (less decline; *p* = 0.02), this difference lost significance after correction for multiple comparisons. No dementia cases occurred, and domain-specific tests (memory, attention, executive function, language) revealed no group differences.
Osipova et al. (2023) [32]	93	Elderly and senile patients with chronic heart failure with low ejection fraction (HFrEF)	OMT for 12 months	MoCA, MMSE	MoCA improved (*p* < 0.05), MMSE correlated with LV EF (r = 0.527, *p* < 0.01), MoCA correlated with LV EF (r = 0.655, *p* < 0.01); MoCA correlated with GFR (r = 0.765, *p* < 0.001), MMSE with GFR (r = 0.671, *p* < 0.001)	Long-term OMT improved myocardial systolic function, GFR, cognitive function, and exercise tolerance. Improvements were more pronounced in elderly patients compared to senile patients.
De Vecchis et al. (2019) [33]	102	Patients with chronic heart failure	Sacubitril/valsartan vs. Standard Therapy	MMSE	Sacubitril/valsartan: 22.72 ± 2.68; Control: 21.96 ± 2.73; *p* = 0.1572	No significant difference in cognitive function between groups; cognitive function remained similar in both groups

Abbreviations: ACEI—angiotensin-converting enzyme inhibitor; ARB—angiotensin receptor blocker; BP—blood pressure; CANTAB—Cambridge Neuropsychological Test Automated Battery; CDR—Clinical Dementia Rating; CI—cognitive impairment/confidence interval; CGNT—Computer-Generated Neuropsychological Tests: reaction time, attention, etc.; DSC—Digit Symbol Coding; GDMT—guideline-directed medical therapy; GFR—Glomerular Filtration Rate; HF—heart failure; HFpEF—heart failure with preserved ejection fraction; HR—hazard ratio; Hs-cTnT—cardiac troponin T measured with a highly sensitive assay; LV EF—Left Ventricular Ejection Fraction; MCI—mild cognitive impairment; MMSE—Mini-Mental State Examination; MoCA—Montreal Cognitive Assessment; NTB—Neuropsychological Test Battery Composite; OMT—Optimal Drug Therapy; NT-proBNP—N-terminal pro-B-type natriuretic peptide; NYHA—New York Heart Association; OR—odds ratio; RCT—randomized controlled trial; TICS-m—modified Telephone Interview for Cognitive Status.

**Table 3 pharmaceuticals-18-01671-t003:** Risk of bias assessment.

Study (Year, Trial Acronym)	Study Design	Tool Used	Result
Haney et al. (2024, SPRINT) [23]	RCT	RoB 2	Low–moderate risk
Dewan et al. (2024, PARAGON-HF) [24]	RCT	RoB 2	Low risk
Palau et al. (2021) [25]	RCT	RoB 2	Low risk
Offer et al. (2019) [26]	RCTs (secondary analysis)	RoB 2	Moderate risk
Bratzke et al. (2016) [27]	Observational	NOS	7/9 stars (moderate)
Hildreth et al. (2015) [28]	RCT	RoB 2	Moderate risk
Johnson et al. (2019) [29]	RCT	RoB 2	Moderate risk
Mavaddat et al. (2014) [30]	RCT	RoB 2	Moderate risk
Caramelli et al. (2022, GIRAF) [31]	RCT	RoB 2	Low risk
Osipova et al. (2023) [32]	Observational	NOS	6/9 stars (moderate)
De Vecchis et al. (2019) [33]	Observational	NOS	7/9 stars (moderate)

Abbreviations: RCT—Randomized Controlled Trial; RoB 2—Cochrane Risk of Bias 2 tool; NOS—Newcastle-Ottawa Scale.

**Table 4 pharmaceuticals-18-01671-t004:** Summary of findings (GRADE).

Intervention/ Exposure	Comparator	Studies (n for Participants)	Main Cognitive Outcome	Effect	Certainty (GRADE)
Intensive BP lowering	Standard BP	1 RCT (n = 8111)	MoCA, MCI/dementia	HR 0.64 (95% CI 0.50–0.81)	Moderate
Sacubitril/ valsartan	RAAS inhibitors	2 studies (1 RCT, n = 2895; 1 observational, n = 102)	MMSE	No difference	Low
Standard HF therapies (β-blockers, ACEI/ARB, diuretics, statins)	Placebo/ usual care	Multiple cohorts + RCTs (n > 45,000)	MMSE, MoCA	No consistent harm	Low
Warfarin	Aspirin	1 RCT (n = 973)	MMSE	No significant difference	Very low
Dabigatran	Warfarin	1 RCT (n = 200)	MMSE, MoCA, NTB	No significant difference after adjustment	Low
Optimized medical therapy (OMT)	Baseline	1 observational (n = 93)	MMSE, MoCA	Cognitive improvement parallel to LV EF and GFR gains	Very low

Abbreviations: BP—blood pressure; RCT—randomized controlled trial; MoCA—Montreal Cognitive Assessment; MCI—mild cognitive impairment; MMSE—Mini-Mental State Examination; NTB—Neuropsychological Test Battery Composite; HR—hazard ratio; CI—confidence interval; RAAS—renin–angiotensin–aldosterone system; ACEI—angiotensin-converting enzyme inhibitor; ARB—angiotensin II receptor blocker; HF—heart failure.

## Data Availability

No new data were created or analyzed in this study. Data sharing is not applicable to this article.

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
