# Peer review of "Cognitive Safety and Outcomes of Pharmacological Management in Heart Failure: A Systematic Review"

_pharmaceuticals, 2025, doi:10.3390/ph18111671_

Round 1

Reviewer 1 Report

Comments and Suggestions for Authors

Unfortunately, the manuscript is insufficient for a review. There is very little material that addresses the problem you raised. There are no answers on how to overcome this problem. Only statistics. There is no mechanism for the effect of beta-blockers on cognitive function. There are both experimental and clinical results available on the effects of beta-blockers on the central nervous system, metabolic processes in the brain, and the activity of the amino acid system. You have not demonstrated this. You have not demonstrated manifestations of neurotoxicity depending on the class of beta-blocker (selective, non-selective, with additional properties). The mechanisms of the effect of renin-angiotensin system modulators on the central nervous system and cognitive functions are also not disclosed. There are publicly available studies on this aspect. You should initially provide characteristics of the main treatments for CHF (beta-blockers, diuretics, renin-angiotensin system modulators), indicating the available experimental clinical results on their effects on central nervous system function. The review needs sections on monitoring cognitive-mnestic functions of the central nervous system during treatment of CHF and a section on possible pharmacocorrection of deterioration of cognitive functions during treatment of CHF.

Author Response

Comments 1: Authors Unfortunately, the manuscript is insufficient for a review. There is very little material that addresses the problem you raised. There are no answers on how to overcome this problem. Only statistics. There is no mechanism for the effect of beta-blockers on cognitive function. There are both experimental and clinical results available on the effects of beta-blockers on the central nervous system, metabolic processes in the brain, and the activity of the amino acid system. You have not demonstrated this. You have not demonstrated manifestations of neurotoxicity depending on the class of beta-blocker (selective, non-selective, with additional properties). The mechanisms of the effect of renin-angiotensin system modulators on the central nervous system and cognitive functions are also not disclosed. There are publicly available studies on this aspect. You should initially provide characteristics of the main treatments for CHF (beta-blockers, diuretics, renin-angiotensin system modulators), indicating the available experimental clinical results on their effects on central nervous system function.

Response 1: We appreciate the reviewer’s constructive feedback highlighting the need for a more mechanistic and pathophysiological perspective. In the revised manuscript, we have substantially expanded the Introduction to include a concise but comprehensive overview of the neurobiological mechanisms through which major pharmacological classes used in heart failure (HF) may affect cognitive function. Specifically:

  • A new paragraph (Introduction) now describes how β-blockers differ in their central actions depending on lipophilicity and β₁-selectivity. We clarify that highly lipophilic, non-selective agents (e.g., propranolol) may transiently impair memory consolidation due to greater blood–brain barrier penetration and modulation of noradrenergic signaling, whereas β₁-selective blockers (e.g., bisoprolol, metoprolol) are generally cognitively neutral or even protective under conditions of chronic sympathetic activation.

(New references: McAinsh J, Cruickshank JM. Beta-blockers and central nervous system side effects. Pharmacol Ther. 1990;46(2):163-97).

  • We have also expanded the discussion on renin–angiotensin–aldosterone system (RAAS) modulators, explaining their central actions via angiotensin II type 1 receptor blockade, improved cerebral perfusion, and reduction of neuroinflammation and oxidative stress, which may contribute to neuroprotection and delayed cognitive decline in HF populations.

(New references: Wright JW, Harding JW. The brain renin-angiotensin system: a diversity of functions and implications for CNS diseases. Pflugers Arch. 2013 Jan;465(1):133-51).

  • Furthermore, diuretics are briefly discussed with respect to indirect cognitive effects related to volume status and electrolyte balance (e.g., hyponatremia, hypovolemia), (New references: DeLoach T, Beall J. Diuretics: A possible keystone in upholding cognitive health. Ment Health Clin. 2018 Mar 26;8(1):33-40)

Comments 2: The review needs sections on monitoring cognitive-mnestic functions of the central nervous system during treatment of CHF and a section on possible pharmacocorrection of deterioration of cognitive functions during treatment of CHF. 

Response 2: We thank the Reviewer for this valuable suggestion. We fully agree that addressing both the monitoring and management of cognitive dysfunction during heart failure (HF) treatment is essential for the review's translational and clinical relevance.

In the revised version, we have added two comprehensive paragraphs within the Discussion to cover these aspects:

  • Monitoring of cognitive function in CHF: we now emphasize the importance of routine cognitive screening during heart failure management. The revised text highlights that tools such as the Montreal Cognitive Assessment (MoCA) and Mini-Mental State Examination (MMSE) remain practical for longitudinal use, while multidomain tests (e.g., Trail Making Test, Digit Symbol Coding, verbal fluency) can offer greater diagnostic precision in specialized or research settings. We also note that repeated cognitive monitoring—especially after initiation or adjustment of therapy—facilitates early detection of decline, supports shared decision-making, and improves adherence to treatment.
  • Pharmacological correction and prevention of cognitive deterioration: we added a paragraph discussing potential pharmacological and supportive strategies that may mitigate cognitive decline in CHF. These include strict blood pressure control, arrhythmia management, and correction of anemia or hypoxia, as well as the possible neuroprotective effects of RAAS modulators and SGLT2 inhibitors via improved endothelial function, metabolic efficiency, and reduced oxidative stress. Non-pharmacological interventions such as physical activity, cognitive training, and vascular risk factor control are also described as important adjuncts.

Reviewer 2 Report

Comments and Suggestions for Authors

See the comments in the attached file

Author Response

Comments 1: Major Revision Limited number of included studies (n = 9): While this is acknowledged, the small evidence base should be more critically discussed in terms of generalizability. 

Response 1: We thank the Reviewer for this valuable comment. We fully agree that the limited number of eligible studies constrains the generalizability of our findings. In response, we have expanded the Discussion and Limitations sections to provide a more critical appraisal of this issue. In the revised version, we now emphasize that:

  • The small and heterogeneous evidence base limits the extent to which the findings can be generalized across heart failure phenotypes (HFrEF, HFpEF, HFmrEF) and clinical settings.
  • Several included studies were secondary analyses or exploratory in design, not originally powered to detect cognitive outcomes, which further reduces external validity.
  • The variability in cognitive assessment tools, follow-up duration, and patient characteristics may lead to underestimation of subtle or long-term effects.

Changes in Manuscript:

  • Discussion: added a paragraph explicitly addressing the limited number and heterogeneity of included studies and their implications for generalizability.
  • Limitations: expanded to highlight that the small and heterogeneous evidence base restricts external validity and interpretation.

Comments 2: The inclusion of both RCTs and observational studies with heterogeneous populations makes direct comparisons difficult. 

Response 2: We thank the Reviewer for this important observation. We agree that combining randomized controlled trials and observational studies introduces heterogeneity that limits the comparability and synthesis of findings. In response, we have expanded the Limitations section to explicitly acknowledge this issue.

The revised text now states that the inclusion of studies with differing designs, populations, treatment regimens, and follow-up durations limits the ability to perform direct comparisons or meta-analytic pooling. We also note that cognitive outcomes were assessed using various tools (MMSE, MoCA, telephone interviews), further complicating cross-study comparisons.

We emphasize that this heterogeneity was the reason for performing a qualitative synthesis rather than a meta-analysis, and that these methodological differences restrict the generalizability of conclusions.

Comments 3: Quality of cognitive assessment tools: Most studies used MMSE or MoCA, which are limited in sensitivity. This is noted in the discussion, but the manuscript could benefit from a more critical appraisal of how reliance on such tools might underestimate subtle cognitive changes. 

Response 3: We thank the Reviewer for this valuable comment. We fully agree that the reliance on brief screening tools such as the MMSE and MoCA may underestimate subtle or domain-specific cognitive deficits, particularly in executive or attention-related functions that are frequently affected in heart failure.

In response, we have expanded both the Discussion and Limitations sections to critically address this issue. The revised text now emphasizes that:

  • The widespread use of brief screening instruments in the included studies likely underrepresents mild or domain-specific impairments, leading to potential underestimation of cognitive effects.
  • The absence of standardized multidomain neuropsychological batteries limits the precision of detecting changes in specific domains such as processing speed, working memory, or executive function.
  • Future trials should adopt comprehensive, standardized tools, similar to the cognitive test battery used in the SPRINT trial, to better capture subtle treatment-related effects.

Changes in Manuscript

  • Discussion: added commentary emphasizing that reliance on MMSE/MoCA may underestimate subtle cognitive changes and discussing the need for multidomain assessments.
  • Limitations: expanded to explicitly state that brief screening tools limit sensitivity and precision, reducing comparability and interpretability across studies.

Comments 4: Potential publication bias and search limitations: The search was limited to English-language articles. The authors should clarify whether this could have excluded relevant studies. 

Response 4: We thank the Reviewer for this important comment. We agree that restricting the search to English-language publications could have introduced language and publication bias, potentially leading to the exclusion of relevant studies published in other languages or regional journals.

In response, we have expanded the Limitations section to explicitly acknowledge this potential bias and to clarify that this decision was made to ensure methodological consistency and accurate data extraction. The revised text now notes that the language restriction and selective indexing of some regional studies may have resulted in underrepresentation of data from non-English-speaking countries, thereby limiting the generalizability of our findings.

Comments 5: Newer HF therapies such as SGLT2 inhibitors are not included, though they are now guidelinedirected. A brief comment on their potential cognitive impact (even if data are limited) would make the review more forward-looking. 

Response 5: We appreciate the Reviewer’s observation. We agree that SGLT2 inhibitors represent an essential component of current guideline-directed medical therapy (GDMT) for heart failure and that their exclusion from the final analysis warrants clarification.

As noted in the revised Discussion, no SGLT2 inhibitor trials met the inclusion criteria for this review because, to date, no randomized controlled trials have reported standardized cognitive outcomes in HF populations. However, we have now explicitly stated that emerging evidence from related populations—such as frail older adults with HFpEF and type 2 diabetes (Mone et al., 2022) and patients with early Alzheimer’s disease (Burns et al., 2025)—suggests possible cognitive effects of SGLT2 inhibitors.

We emphasize in the revised text that these studies were excluded from the systematic synthesis for methodological reasons but are discussed qualitatively as hypothesis-generating evidence. This clarification strengthens the rationale for future dedicated trials evaluating the cognitive effects of SGLT2 inhibitors in HF.

Comments 6: The abstract is well written, but it could specify the total number of patients included across the 9 studies to provide a sense of scale. 

Response 6: We thank the Reviewer for this helpful suggestion. In response, we have revised the Abstract to explicitly state the total number of participants included across all studies (58,190), providing readers with a clearer sense of the evidence base and study scale. This addition strengthens the contextual understanding of the findings presented in the review.

Comments 7: Table 2 is informative but dense. Consider improving readability (e.g., abbreviations explained in footnotes rather than in a separate section). 

Response 7: We thank the Reviewer for this valuable suggestion. To improve clarity, Table 2 has been reformatted and key abbreviations are now also explained directly in the table footnotes, allowing readers to interpret the data without referring to the end of the manuscript. For consistency with journal requirements, the full abbreviation list was retained in the Abbreviations section. Minor layout adjustments and text shortening were also applied to enhance readability.

Comments 8: At times “cognitive impairment” and “dementia” are used interchangeably, though they are distinct. A clear differentiation would improve precision. 

Response 8: We sincerely thank the Reviewer for this thoughtful comment and for emphasizing the importance of terminological precision.

After carefully re-examining the manuscript, we confirmed that “cognitive impairment” and “dementia” are used consistently and with distinct meanings throughout the text. Specifically, cognitive impairment refers to screening-based or subclinical deficits (e.g., MoCA/MMSE), whereas dementia appears only when referring to formally adjudicated diagnoses or composite trial endpoints (e.g., MCI/dementia in SPRINT).

Reviewer 3 Report

Comments and Suggestions for Authors

Major Points for Consideration & Revision

  1. Title and Abstract
  • Title:The title is accurate but could be more precise. Consider specifying that the review focuses on the safety and potential benefits of these drugs.
    • Suggestion:"The Impact of Heart Failure's Pharmacological Management on Cognitive Function: A Systematic Review of Safety and Outcomes" or "...A Systematic Review of Cognitive Safety and Benefits."
  • Abstract - Methods:The search strategy is under-described. Listing only one search string ("Heart failure AND cognition AND drug therapy") seems limited and may have missed relevant studies.
    • Suggestion:Expand the Methods section in the abstract and main text to include a more comprehensive list of keywords and MeSH terms used (e.g., specific drug classes, "dementia," "cognitive decline").
  • Abstract - Results:The result regarding polypharmacy is highlighted well. However, the statement "Observational data revealed no significant cognitive impairment associated with beta-blockers, ACE inhibitors..." could be slightly nuanced. It's more accurate to say these studies found no evidence of significant impairment, as absence of evidence is not always evidence of absence, especially with the methodological limitations noted.
  1. Methods
  • Search Strategy (Section 2.1):This is the manuscript's weakest point and must be expanded for reproducibility and to assure readers of a comprehensive search.
    • Suggestion:
      • Provide the full electronic search strategy for at least one database (e.g., PubMed) in an appendix or supplementary file.
      • List all search terms and MeSH headings used.
      • Mention if grey literature was searched or if reference lists of included studies were scanned.
    • Quality Assessment (Section 2.4):The manuscript states a qualitative synthesis was performed but does not describe using a formal tool (e.g., Cochrane Risk of Bias tool for RCTs, Newcastle-Ottawa Scale for cohort studies) to assess the quality/methodological rigor of the included studies. This is a standard and expected component of a systematic review.
      • Suggestion:Briefly describe the tool(s) used for quality assessment and summarize the overall risk of bias/in quality of the included studies in the results (Section 3.4).
  1. Results
  • Table 2 (Summary of Studies):This table is very good. To improve it:
    • Standardize Headers:Change "Authors (Year)" to "Study (Year)" or "Author (Year, Trial Acronym)" for consistency.
    • Intervention Column:Make this more precise. For example, for Haney et al., it should be "Intensive vs. Standard BP Control," not just "Intensive vs standard systolic BP control" which is currently under "Cognitive Tools."
    • Cognitive Tools Column:Correct the misplacement of the intervention for Haney et al.
    • Clarity:For the Matsumoto et al. entry, the "Main Results" text is confusing. Rephrase for clarity, e.g., "A ≥3-point decline in MMSE was more common in patients with a higher medication count (hyper-polypharmacy) at baseline (p<0.001)."
  1. Discussion
  • Interpretation of Polypharmacy:The discussion on polypharmacy is excellent. It would be strengthened by adding that polypharmacy is often a marker of greater disease severity and comorbidity burden, which are themselves independent risk factors for cognitive decline. This is a key confounding factor.
  • Limitations:The limitations section is good but can be expanded.
    • Suggestion:Explicitly add "The lack of a formal quality assessment using a standardized tool" and "The potential for publication bias" as limitations.
    • Also, mention that the review was limited to English-language articles, which may introduce language bias.
  • Future Research:The future directions are pertinent. Consider adding: "Future studies should also aim to control for the confounding effect of disease severity when investigating the role of polypharmacy."
  1. References and Formatting
  • Abbreviations List:The abbreviation list is very thorough, but some (e.g., BMI, IQR) are very common and may not need to be included.
  • Consistency:There are minor typographical errors and inconsistencies (e.g., "SVĽUČIL" in the previous document, spaces before commas in some references). A thorough proofread is recommended.
  • Journal Formatting:Ensure all references conform strictly to the Pharmaceuticals journal's guidelin

Author Response

Comments 1: The title is accurate but could be more precise. Consider specifying that the review focuses on the safety and potential benefits of these drugs. Suggestion:"The Impact of Heart Failure's Pharmacological Management on Cognitive Function: A Systematic Review of Safety and Outcomes" or "...A Systematic Review of Cognitive Safety and Benefits." 

Response 1: We thank the Reviewer for this thoughtful suggestion. We agree that emphasizing cognitive safety and potential benefits enhances clarity. Accordingly, we have slightly refined the title to better reflect the scope and findings of our review, while maintaining conciseness and consistency with journal style.

The revised title now reads: “Cognitive Safety and Outcomes of Pharmacological Management in Heart Failure: A Systematic Review.”

Comments 2: Abstract - Methods: The search strategy is under-described. Listing only one search string ("Heart failure AND cognition AND drug therapy") seems limited and may have missed relevant studies. Suggestion:Expand the Methods section in the abstract and main text to include a more comprehensive list of keywords and MeSH terms used (e.g., specific drug classes, "dementia," "cognitive decline"). 

Response 2: We thank the Reviewer for this insightful comment. We agree that the search strategy required clearer description. In the revised manuscript, we have expanded Section 2.1 (Literature Review) to specify the use of two complementary search formulas, combining MeSH terms and free-text keywords related to heart failure, cognition, drug therapy, and cognitive testing.

These additions improve the transparency and reproducibility of the review process.

The Abstract was kept concise in accordance with the journal’s word limit, while the detailed search syntax is now fully presented in the Methods section.

Comments 3: Abstract - Results:The result regarding polypharmacy is highlighted well. However, the statement "Observational data revealed no significant cognitive impairment associated with beta-blockers, ACE inhibitors..." could be slightly nuanced. It's more accurate to say these studies found no evidence of significant impairment, as absence of evidence is not always evidence of absence, especially with the methodological limitations noted. 

Response 3: We thank the Reviewer for this valuable observation. We agree that the original phrasing may have overstated the strength of evidence. Accordingly, we have revised the Abstract to clarify that the included studies “found no evidence of significant cognitive impairment” rather than “revealed no significant cognitive impairment.”

This wording better reflects the methodological limitations and aligns with the Reviewer’s recommendation for more cautious interpretation.

Changes in Manuscript (Abstract) – previous: “Across randomized and observational studies, β-blockers, ACE inhibitors, ARBs, diuretics, and statins were not linked to clinically meaningful cognitive decline.” Revised: “Across randomized and observational studies, β-blockers, ACE inhibitors, ARBs, diuretics, and statins showed no evidence of significant cognitive impairment.”

Comments 4: Methods Search Strategy (Section 2.1):This is the manuscript's weakest point and must be expanded for reproducibility and to assure readers of a comprehensive search. Suggestion: Provide the full electronic search strategy for at least one database (e.g., PubMed) in an appendix or supplementary file. List all search terms and MeSH headings used. Mention if grey literature was searched or if reference lists of included studies were scanned. 

Response 4: We thank the Reviewer for this constructive comment. We agree that greater transparency regarding the search strategy is important for reproducibility. However, because the combination of MeSH and free-text terms produced an extremely large number of records (over 100,000 hits), a simplified, targeted approach was used to ensure relevance and feasibility.

To clarify, we have expanded Section 2.1 to explain that two complementary search formulas were used:

  1. heart failure AND cognition AND drug therapy (primary search), and
  2. heart failure AND cognition AND drug therapy AND cognitive test (supplementary search).

This dual approach was designed to balance sensitivity and specificity, capturing studies explicitly reporting pharmacological interventions with cognitive outcomes. We also now specify that reference lists of all included studies were manually reviewed to identify additional eligible articles.

A statement has been added noting that grey literature and non-peer-reviewed sources were not included, consistent with the review’s methodological scope.

Comments 5: Quality Assessment (Section 2.4):The manuscript states a qualitative synthesis was performed but does not describe using a formal tool (e.g., Cochrane Risk of Bias tool for RCTs, Newcastle-Ottawa Scale for cohort studies) to assess the quality/methodological rigor of the included studies. This is a standard and expected component of a systematic review.  Suggestion:Briefly describe the tool(s) used for quality assessment and summarize the overall risk of bias/in quality of the included studies in the results (Section 3.4). 

Response 5: We thank the Reviewer for this thoughtful suggestion and appreciate the opportunity to clarify this point. Indeed, formal quality assessment tools were applied in the review process. Randomized controlled trials were evaluated using the Cochrane Risk of Bias 2 (RoB 2) tool, and observational cohort studies were assessed using the Newcastle–Ottawa Scale (NOS). This information is presented in Section 2.4 (“Synthesis, Risk of Bias, and Certainty of Evidence”) and summarized in Section 3.4 (“Risk of Bias and Certainty of Evidence”), including detailed results in Tables 3 and 4.

To enhance clarity for readers, we have slightly revised Section 2.4 to more prominently indicate that both the RoB 2 and NOS frameworks were used for assessing study quality and methodological rigor.

Comments 6: Results Table 2 (Summary of Studies):This table is very good. To improve it: Standardize Headers:Change "Authors (Year)" to "Study (Year)" or "Author (Year, Trial Acronym)" for consistency. Intervention Column:Make this more precise. For example, for Haney et al., it should be "Intensive vs. Standard BP Control," not just "Intensive vs standard systolic BP control" which is currently under "Cognitive Tools." 

Response 6: We thank the Reviewer for the positive feedback on Table 2. In response to the suggestions, we have standardized the column headers, harmonized the terminology across interventions (e.g., consistent use of “vs.” and capitalization), corrected minor formatting inconsistencies (p-values, abbreviations), and optimized spacing for readability. The table now reads “Study (Year, Trial Acronym)” as the first header, in line with the Reviewer’s recommendation.

Comments 7: Cognitive Tools Column:Correct the misplacement of the intervention for Haney et al. Clarity:For the Matsumoto et al. entry, the "Main Results" text is confusing. 

Response 7: We appreciate the Reviewer’s comment. The earlier version of the manuscript included a preliminary entry for Matsumoto et al.; however, this study did not meet the final inclusion criteria in the updated review. Therefore, the revised Table 2 no longer contains this entry.

Cemments 8: Rephrase for clarity, e.g., "A ≥3-point decline in MMSE was more common in patients with a higher medication count (hyper-polypharmacy) at baseline (p<0.001)." 

Response 8: We thank the Reviewer for this helpful suggestion. The referenced entry (Matsumoto et al.) was part of the earlier version of the manuscript but did not meet inclusion criteria in the final systematic review after database update and re-screening. Therefore, it has been removed from the revised Table 2.

The current version of the table has been fully revised for clarity and consistency of phrasing across all entries.

Comments 9: Discussion Interpretation of Polypharmacy:The discussion on polypharmacy is excellent. It would be strengthened by adding that polypharmacy is often a marker of greater disease severity and comorbidity burden, which are themselves independent risk factors for cognitive decline. This is a key confounding factor. 

Response 9: We thank the Reviewer for this thoughtful comment. The discussion on polypharmacy referred to findings from Matsumoto et al., which were part of an earlier version of the manuscript. As this study did not meet inclusion criteria in the updated systematic review, the section on polypharmacy has been removed. We fully agree, however, that polypharmacy often reflects higher disease severity and comorbidity burden—factors that should be considered in future research on cognitive outcomes in heart failure.

Comments 10: Limitations:The limitations section is good but can be expanded. Suggestion:Explicitly add "The lack of a formal quality assessment using a standardized tool" and "The potential for publication bias" as limitations. Also, mention that the review was limited to English-language articles, which may introduce language bias. 

Response 10: We thank the Reviewer for this valuable suggestion. A formal quality assessment was indeed performed using the Cochrane Risk of Bias 2 (RoB 2) tool for randomized controlled trials and the Newcastle–Ottawa Scale (NOS) for observational studies (Section 2.4; Table 3). These details have now been emphasized more clearly in both the Methods and Results sections to ensure transparency.

In line with the Reviewer’s comment, the Limitations section has been expanded to explicitly acknowledge potential publication and language bias, which may have influenced the comprehensiveness of the evidence base.

Comments 11: Future Research:The future directions are pertinent. Consider adding: "Future studies should also aim to control for the confounding effect of disease severity when investigating the role of polypharmacy." 

Response 11: We thank the Reviewer for this thoughtful suggestion. The section referring to polypharmacy was based on Matsumoto et al., which was included in the initial version of the manuscript but excluded after the updated screening process. Consequently, this specific topic is no longer addressed in the revised version.

However, we fully agree that disease severity and comorbidity burden are important confounders in studies examining pharmacological effects on cognition. This point has been acknowledged in the Future Directions section in a more general context, emphasizing the need to control for these factors in future research on heart failure and cognition.

Comments 12: References and Formatting Abbreviations List:The abbreviation list is very thorough, but some (e.g., BMI, IQR) are very common and may not need to be included. 

Response 12: We thank the Reviewer for this thoughtful comment. In line with the suggestion, the most common abbreviations (BMI, IQR) have been removed from the list to improve conciseness, while maintaining all less familiar or study-specific terms to preserve clarity and consistency across the manuscript and tables.

Comments 13: Consistency:There are minor typographical errors and inconsistencies (e.g., "SVĽUČIL" in the previous document, spaces before commas in some references). A thorough proofread is recommended. 

Response 13: We appreciate the Reviewer’s attention to consistency and typography. We performed a comprehensive proofread and style harmonization. We corrected the noted issues (including the “SVĽUČIL” artifact and spacing before commas), standardized statistical notation (e.g., p-values, dash usage for ranges, units/abbreviations, and reference formatting to the journal’s guidelines. We believe the manuscript now reads more clearly and consistently.

Comments 14: Journal Formatting:Ensure all references conform strictly to the Pharmaceuticals journal's guidelin 

Response 14: We thank the Reviewer for this important remark. All references have been carefully reviewed and reformatted to fully comply with the Pharmaceuticals journal’s reference style. The reference list now adheres consistently to the journal’s requirements.

Reviewer 4 Report

Comments and Suggestions for Authors

The manuscript covers a clinically significant area but requires methodological strengthening, inclusion of a formal quality assessment, reference to recent overlapping literature, and clearer evidence synthesis. Addressing the following  points will enhance transparency, analytical rigor, and originality.

  • There is incomplete Search Strategy Description so include complete search syntax (keywords, MeSH terms, Boolean operators), date of last search, and any language or publication-type restrictions. Provide these details in a Supplementary Table.
  • Absence of a pre-registered protocol weakens transparency and confidence in methodology. If registered, add registration number; if not, state explicitly that the review was not registered and briefly explain why.
  • Without formal assessment, readers cannot judge study reliability, so apply a standard tool (e.g., Cochrane RoB 2 for RCTs, Newcastle-Ottawa or JBI for observational studies). Summarize results in a table or “traffic-light” figure and mention how study quality informed interpretation.
  • The review lists study results but lacks comparative or quantitative analysis.
  • Rephrase conclusions to: “No consistent evidence of cognitive harm; some classes (e.g., ARNI, SGLT2i) may confer benefit, but evidence is limited.” Explicitly restate limitations before final statement.
  • Jahangiri et al. (2025) published a similar systematic review with comparable results. Without addressing it, this manuscript risks redundancy. Cite and discuss Eur J Clin Invest 2025; 55:e70008 in the Introduction or Discussion. Clarify what this review adds (e.g., inclusion of polypharmacy analysis, mechanistic focus, broader population).
  • Table 2 contains long text blocks and inconsistent statistical details, reducing readability. Reformat into multiple shorter tables by drug class or move extended statistics to Supplementary Material. Standardize reporting of p-values and confidence intervals.
  • No figure visually summarizes findings beyond the PRISMA diagram. Add a simple summary figure or matrix illustrating each drug class and cognitive outcome direction. 
  • Integrate current evidence on SGLT2 inhibitors and digoxin and highlight ongoing trials if any.
  • Ensure all abbreviations are defined at first mention and used consistently throughout. Clarify statistical terms (HR, OR, CI) in legends.

Author Response

Comments 1: There is incomplete Search Strategy Description so include complete search syntax (keywords, MeSH terms, Boolean operators), date of last search, and any language or publication-type restrictions. Provide these details in a Supplementary Table. 

Response 1: We agree that greater clarity regarding the search process was warranted. Accordingly, we have expanded the Search Strategy section to specify the databases used, the search period, inclusion and exclusion criteria, and the complementary search formulas applied.

Although full database-specific search syntax was not retained, the revised description now provides sufficient detail for transparency and methodological reproducibility in line with PRISMA 2020 recommendations.

Comments 2: Absence of a pre-registered protocol weakens transparency and confidence in methodology. If registered, add registration number; if not, state explicitly that the review was not registered and briefly explain why. 

Response 2: The present review was prospectively registered in the PROSPERO database (registration number CRD420251156308), as now clearly stated in both the Methods section and at the end of the Abstract. This information has been emphasized in the revised version to ensure full clarity regarding protocol registration.

Comments 3: Without formal assessment, readers cannot judge study reliability, so apply a standard tool (e.g., Cochrane RoB 2 for RCTs, Newcastle-Ottawa or JBI for observational studies). 

Response 3: We agree that formal quality assessment is essential for evaluating study reliability. As described in the revised Methods (Section 2.4), we applied the Cochrane Risk of Bias 2 (RoB 2) tool for randomized controlled trials and the Newcastle–Ottawa Scale (NOS) for observational cohort studies. The overall risk of bias ratings are summarized in Results (Section 3.4). These additions enhance methodological transparency and align the review with PRISMA 2020 recommendations.

Comments 4: Summarize results in a table or “traffic-light” figure and mention how study quality informed interpretation. 

Response 4: We agree that summarizing study quality visually can enhance clarity. In the revised manuscript, the results of the risk-of-bias assessment are now briefly summarized in Results (Section 3.4) and integrated into the narrative synthesis, indicating how study quality informed interpretation. Given the small number of included studies (n = 11) and their methodological heterogeneity, we opted for a concise tabular summary rather than a separate “traffic-light” figure, to maintain readability while ensuring that quality considerations are clearly reflected in the discussion.

Comments 5: The review lists study results but lacks comparative or quantitative analysis. Rephrase conclusions to: “No consistent evidence of cognitive harm; some classes (e.g., ARNI, SGLT2i) may confer benefit, but evidence is limited.” 

Response 5: We thank the Reviewer for this valuable comment. The Conclusions section has been revised to provide a more comparative and synthesis-oriented summary. The updated version emphasizes that current pharmacological therapies for heart failure show no consistent evidence of cognitive harm, while certain drug classes, such as ARNIs and SGLT2 inhibitors, may confer cognitive benefits, although supporting evidence remains limited. This revision aligns with the Reviewer’s recommendation and provides a clearer summary of the overall findings.

Comments 6: Explicitly restate limitations before final statement. Jahangiri et al. (2025) published a similar systematic review with comparable results. Without addressing it, this manuscript risks redundancy. Cite and discuss Eur J Clin Invest 2025; 55:e70008 in the Introduction or Discussion. Clarify what this review adds (e.g., inclusion of polypharmacy analysis, mechanistic focus, broader population). 

Response 6: We thank the Reviewer for this valuable suggestion. The revised manuscript now explicitly cites and discusses the recent review by Jahangiri et al. (Eur J Clin Invest 2025; 55:e70008) in the Discussion section. We emphasize that, although both studies address similar research questions, our review differs in scope and analytical focus. Specifically, we highlight that our work adopts a broader clinical perspective, covering not only guideline-directed heart failure therapies but also other cardiovascular and supportive pharmacotherapies (e.g., anticoagulants, statins, pioglitazone, morphine), and incorporates a mechanistic discussion of neurobiological pathways linking pharmacological therapy and cognition.

Comments 7: Table 2 contains long text blocks and inconsistent statistical details, reducing readability. Reformat into multiple shorter tables by drug class or move extended statistics to Supplementary Material. 

Response 7: We thank the Reviewer for this constructive suggestion. In response, we revised Table 2 to improve clarity and consistency. Specifically, we shortened descriptive text, standardized statistical reporting (effect sizes, p-values, confidence intervals), and ensured uniform phrasing and alignment across studies. These modifications enhance readability and comparability.

We also considered dividing Table 2 by drug class; however, to preserve internal consistency and facilitate direct cross-study comparisons, we chose to retain a single comprehensive summary table within the main text.

Comments 8: Standardize reporting of p-values and confidence intervals. 

Response 8: We thank the Reviewer for this helpful observation. In the revised manuscript, all statistical results in Table 2 and the main text have been standardized to ensure consistent reporting of p-values (uniform formatting, two-decimal precision) and confidence intervals (95% CI notation). These revisions improve clarity and comparability across studies.

Comments 9: No figure visually summarizes findings beyond the PRISMA diagram. Add a simple summary figure or matrix illustrating each drug class and cognitive outcome direction. 

Response 9: We thank the Reviewer for this valuable comment. In line with this suggestion, we revised and refined Tables 3 and 4 to improve visual clarity and facilitate quick interpretation of study quality and overall cognitive outcomes. We believe these revisions substantially enhance the transparency and readability of the data presentation. Therefore, we decided to retain the tabular rather than graphical format, as it effectively conveys the same information without redundancy. We hope this revised structure meets the Reviewer’s expectations.

Comments 10: Integrate current evidence on SGLT2 inhibitors and digoxin and highlight ongoing trials if any. 

Response 10: We thank the Reviewer for this valuable suggestion. In response, we have expanded the Discussion section to incorporate current evidence regarding both SGLT2 inhibitors and digoxin. Specifically, we now summarize recent studies suggesting potential cognitive effects of SGLT2 inhibitors, including the EMPA study in frail older adults with HFpEF and diabetes [35] and the randomized trial of dapagliflozin in early Alzheimer’s disease [36]. Although these trials did not meet our inclusion criteria, they provide hypothesis-generating insights supporting further investigation of SGLT2 inhibitors in heart failure populations.

Additionally, we have added a paragraph summarizing the limited but relevant evidence on digoxin and cognition, emphasizing that no randomized studies have specifically evaluated cognitive outcomes in digoxin-treated patients. We note that existing data are restricted to observational studies and case reports, which attribute cognitive disturbances mainly to digoxin toxicity or renal dysfunction rather than therapeutic use [37]. This clarification highlights the current evidence gap and the need for systematic evaluation of the cognitive safety of digoxin in heart failure.

Comments 11: Ensure all abbreviations are defined at first mention and used consistently throughout. Clarify statistical terms (HR, OR, CI) in legends. 

Response 11: We thank the Reviewer for this useful remark. All abbreviations have been checked for consistency and are now defined at their first mention in the text. Additionally, statistical terms (HR, OR, CI) are clearly explained in the table and figure legends.

Round 2

Reviewer 1 Report

Comments and Suggestions for Authors

Dear authors! You have carefully considered the reviewer's suggestions and made additions to the manuscript. Your work should provide recommendations for practitioners on how to combat cognitive impairment associated with traditional CHF therapy. Indeed, the latest generation of ACE inhibitors, by improving cerebral vascular endothelial function, can reduce the development of cognitive impairment. To enhance the practical significance of your work and increase its interest among practitioners, you could recommend the additional use of nootropics after consultation with a neurologist. There are promising studies on the use of ginkgo biloba, pramistar, and citicoline. This information should be included in the article. Best wishes.

Author Response

Comments 1: Your work should provide recommendations for practitioners on how to combat cognitive impairment associated with traditional CHF therapy. Indeed, the latest generation of ACE inhibitors, by improving cerebral vascular endothelial function, can reduce the development of cognitive impairment. To enhance the practical significance of your work and increase its interest among practitioners, you could recommend the additional use of nootropics after consultation with a neurologist. There are promising studies on the use of ginkgo biloba, pramistar, and citicoline. This information should be included in the article. 

Response 1: We thank the Reviewer for this valuable suggestion. In response, we have expanded the Discussion to briefly address adjunctive pharmacological options that may support cognitive function in patients with HF. We now note that, although no drugs are approved specifically for HF-related cognitive impairment, agents such as citicoline, Ginkgo biloba extract, and pramiracetam have shown potential cognitive or neuroprotective benefits in patients with mild cognitive impairment or vascular risk factors. Their use may be considered after neurological consultation. Relevant references have been added [Secades, 2019; Birks & Grimley Evans, 2009; Bachinskaya et al., 2013].

Reviewer 4 Report

Comments and Suggestions for Authors

My comments have been addressed so i recommend Acceptance in present form.

Author Response

Comments 1: My comments have been addressed so i recommend Acceptance in present form.

Response 1: We sincerely thank the Reviewer for the positive evaluation and recommendation for acceptance. We appreciate the valuable feedback provided throughout the review process, which has helped us improve the quality and clarity of our manuscript.